# Agricultural Impacts on Hydrobiogeochemical Cycling in the Amazon: Is There Any Solution?

**Ricardo de O. Figueiredo** [1,*], **Anthony Cak** [2] **and Daniel Markewitz** [3]

[1]   Embrapa Environment Embrapa Environment, Brazilian Agricultural Research Corporation, Ministry of Agriculture, Livestock, and Food Supply, Jaguariuna 13918-110, Brazil

[2]   Environmental Sciences Initiative, CUNY Advanced Science Research Center, New York, NY 10031 USA; acak@gc.cuny.edu

[3]   Warnell School of Forestry and Natural Resources, The University of Georgia, Athens, GA 30602, USA; dmarke@uga.edu

*   Correspondence: ricardo.figueiredo@embrapa.br; Tel.: +55-19-99717-0087

**Abstract:** Expansion of agriculture in the Brazilian Amazon has been driven not just by demands from traditional, rural producers, but also large agriculture and cattle producers, both of whom have put considerable pressure on remaining forests and their watersheds. Monitoring of these watersheds has been a focus of intensive study for the past 20 years and although this work has greatly increased our understanding, considerable gaps still remain in our ability to provide adequate recommendations for land management and associated public policies. In this study we present a summary of findings from these previous results. For small properties, the use of fire to prepare land for cultivation remains controversial, while in large properties, forest conversion to pasture and/or crop production has had a meaningful and adverse effect on water quality. Riparian forest conservation can make a significant difference in reducing impacts of land-use change. Secondary vegetation can also play an important role in mitigating these impacts. New types of sustainable agricultural production systems, together with incentives such as payments for ecosystem service can also contribute. Continued monitoring of these changes, together with robust sustainable development plans, can help to preserve forest while still addressing the social and economic needs of Amazonian riverine inhabitants.

**Keywords:** biogeochemistry; deforestation; land management; land-use change; public policies; water resources; watersheds

## 1. Introduction

Avoiding deforestation and enforcing restoration to mitigate natural resource degradation is a planetary need, driven in large part by the actions of the agribusiness sector in response to the global demand for food, fiber, and biofuels. Unprecedented increases in demand for these products has led to considerable decreases in the supply of available land [1]. A sustainable future requires not just a dependence on restoring or maintaining forests but also new ways to respond to planetary needs with reduced impacts. While deforestation is a global problem, it has been a particularly important issue in the Amazon, especially the Brazilian Amazon [2], where the "Arc of Deforestation" has been the world's most active deforestation frontier in recent decades [1]. In this region, there is considerable tension between enforcing restoration through public forest protection policies and market factors (e.g., the world price for soybean and beef), often with severe consequences [3].

Agricultural development in the Brazilian Amazon over the past 50 years began chiefly from colonist and settler policies and programs attracting small, traditional producers to the region [4]. Since then, this development has continued and has been joined by large increases in pasture and cropland

area from agribusiness, often into previously deforested areas. While small farmers contribute to the majority of deforestation events, they are responsible for only a fraction of the total deforested area [4]. In terms of overall magnitude, therefore, impacts by agribusiness are much larger and are projected to continue to grow, with expanding pasture to supply the beef market and cropland to produce soybean, along with the development of new varieties of soybean and infrastructure for processing and transport [5].

Continued expansion of agriculture comes at the cost of land-use intensification; the current productivity of Brazilian cultivated pasturelands is 32–34% of its potential, while increasing productivity to 49–52% of the potential would suffice to meet demands for meat, crops, wood products and biofuels until at least 2040 without further conversion of natural ecosystems [6]. Moreover, there is a considerable challenge to constructing a strategy for promoting a new model of rural development in the Amazon, particularly one that includes positive incentives for landholders, indigenous communities, counties, and states to make the transition to low deforestation, productive, sustainable development [7]. Currently, however, agricultural production continues to expand, with potential impacts to climate and greenhouse-gas emissions, and to the cycling of carbon, nitrogen, and water [8].

Agricultural development has greatly impacted the biogeochemical cycling of nutrients and organic and inorganic materials in rivers and streams, which has been shown to be intricately associated with processes operating in adjoining riparian and upland ecosystems, such that the natural flow of Amazon stream water emerges directly from the extensive forests and savannas that comprise the basin [9]. When deforestation occurs (e.g., forest conversion to pasture and then pasture to crops), hydrological watershed processes also respond, often with increased discharge and nutrients in stream and river water. Such impacts are particularly acute in first-order streams where there is no upstream input and all water, particulates, and solutes derive from areas immediately adjacent to the stream [9,10]. However, the effects of land-use change on nutrient export to aquatic ecosystems and the atmosphere must be understood within the context of stream orders as well as varying soil properties across the Amazon Basin [11].

Considering the scale and spatial diversity of the basin, Richey et al. [12] proposed an approach that examines river corridors at three scales: small watersheds (<10 km$^2$), mesoscale areas (~10,000 km$^2$), and the whole basin (7 × 106 km$^2$), each taking into account linkages between water bodies and their surrounding landscapes through biogeochemical and hydrological cycles. Since this suggested approach, an increasing number of studies have added to the body of literature demonstrating negative impacts of land use on the biogeochemistry and hydrology of the Amazon across different scales. Oftentimes, these studies have compared paired or a group of small watersheds, sampling watershed outlet points as a means of assessing upstream impacts to watershed biogeochemistry, while other studies have used upstream-downstream sampling approaches focused on a targeted sector of the catchment [13].

Coupled with these field surveys has been an increasing use and development of models that synthesize existing environmental data to predict hydrologic and biogeochemical fluxes. Despite the relative abundance of field studies over the past 20 years, considerable gaps in data (both availability and quality) have hindered the full development and deployment of such models [14], and consequently, the ability for these predictive models to be used in the development of strategies for sustainable management of the Amazon's forest, soils, and fresh water. In this review, we present and discuss an overview of watershed biogeochemistry research in the Amazon to date, particularly work developed over the last 20 years that has focused on the relationships of biogeochemistry to land and water management by smallholders and large farmers. We also provide recommendations to address both model gaps and public policy for land management in the agriculture frontiers in the Amazon.

## 2. Scientific Works Developed During the Last 20 Years in Different Amazonian Regions

### 2.1. Location of Studied Watersheds

A comprehensive literature review of works was gathered from the authors' publication database archive as well as from literature searches using Google Scholar. While not an exhaustive or complete list of all available works, this review focuses on research on biogeochemical changes driven by either smallholders or large farm activities published since 1997. In total, over 180 references were consulted (full list of references found in the Supplemental Document). When considering studies on biogeochemistry and water flow, we identified that most work to date has been focused along the major rivers of the Amazon, particularly the Amazon mainstem into the Negro and Solimões rivers, as well as the Madeira River (Figure 1). Studies of smaller watersheds have primarily occurred in the Brazilian state of Rondônia, along the Transamazon (BR-230) and Cuiabá-Santarém (BR-163) highways, which cut across the center (east-west and north-south, respectively) of the basin. Additionally, concentrated areas of focus have occurred in the northeastern Brazilian state of Pará, near its capital Belém at the mouth of the Amazon River, and scattered throughout the Brazilian states of Mato Grosso, Roraima and Acre. A lesser number of studies were conducted in Amazonian areas outside of Brazil, mainly in a few sites in Peru, Bolivia, and Ecuador. Surprisingly, the number and location of studies of stream and river hydrobiogeochemistry do not entirely correspond with the 'Arc of Deforestation' of the south and the southeastern part of the Amazon Basin (except for the studies in Rondônia and the few in Mato Grosso), indicating that areas undergoing the most changes may not be the ones that are most studied.

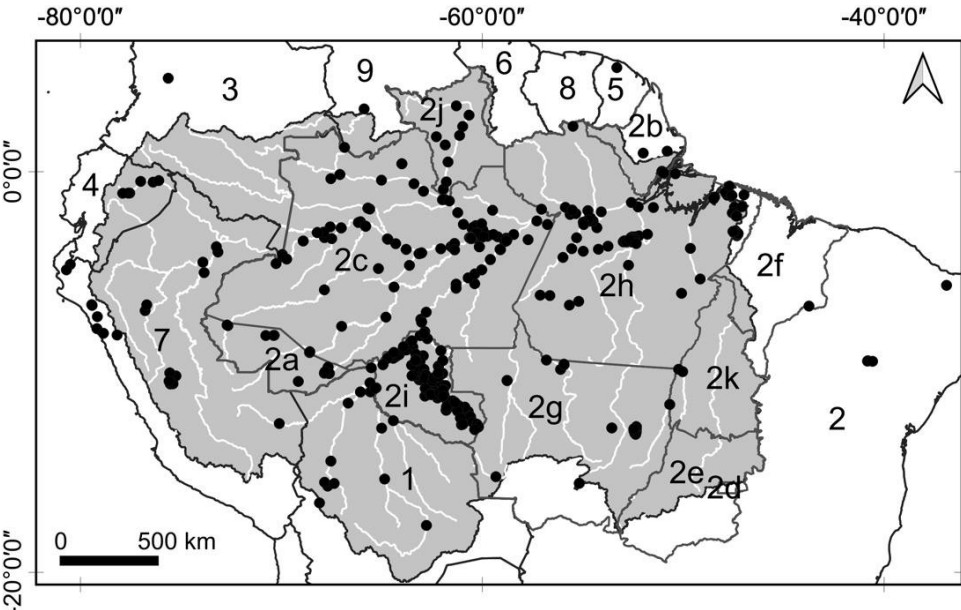

**Figure 1.** Approximate locations (black circles) of sampling areas of studies that assess hydrobiogeochemical aspects of streams and rivers in and near the Amazon Basin (n = 133). Amazon River watershed is shown in gray, with major rivers shown in white. Amazonian countries and Brazilian states included in the Amazon River watershed are identified by numbers/letters: 1: Bolivia; 2: Brazil and states (2a: Acre; 2b: Amapá; 2c: Amazonas; 2d: Distrito Federal; 2e: Goiás; 2f: Maranhão; 2g: Mato Grosso; 2h: Pará; 2i: Rondônia; 2j: Roraima; and 2k: Tocantins); 3: Colombia; 4: Ecuador; 5: French Guiana; 6: Guyana; 7: Peru; 8: Suriname; and 9: Venezuela. List of studies included in the Supplemental Document.

### 2.2. Watershed Biogeochemistry Affected by Small Farming

While large-scale agriculture is responsible for most deforestation in the Amazon, small farmers play an important role in forest management through the formation of a complex social-environmental

mosaic in the region [4]. As they represent an important group driving land cover change in the region, their activities can affect stream water of small watersheds, particularly in areas of colonization settlements (for agrarian reform projects) [15].

Factors such as household ages have been shown to drive stream water biogeochemistry responses, largely through land use choices, as revealed by a study of 25 watersheds in the region near Altamira, state of Pará, along the Transamazon Highway and close to the Xingu River. For example, older, more established households managed properties with abundant areas of forest and agroforestry activities such as cocoa production, leading to streams with 'improved' stream water quality, while both the "middle-age" and youngest households typically engaged in cattle ranching and pasture areas and were associated with streams with degraded water quality (higher concentrations of many ions, especially nitrogen and phosphorus, increased stream water temperature, and decreased dissolved oxygen concentrations) [15].

Typical land-use patterns of smallholders in the Amazon involve the use of fallow periods in the agricultural planting cycle, resulting in forest regrowth. During this secondary vegetation regrowth, throughfall and stemflow under the canopy can provide nutrient inputs, such as nitrogen, potassium, calcium, magnesium, and sulfur, to soils with rates varying by species composition [16]. As farmers frequently use fire to prepare the land for cultivation, secondary vegetation is burned in the dry season, resulting in ash deposition on the canopy and subsequent wash-off and solution of ash particles in stream and river water [17]. This scenario has been observed in many parts of the Amazon, including the "Zona Bragantina" of the northeastern part of the Brazilian state of Pará. However, alternatives to using fire have been explored, especially through sequential agroforestry, prescribed secondary vegetation management, and chop-and-mulch land management [18]. Nutrient and water losses from soils can be avoided with chop-and-mulch; despite the relatively higher amount of rapidly decomposing surface mulch compared to slash-and-burn scenarios, this does not increase nutrient losses by leaching [19].

In this alternative management context, a study was conducted on three first order catchments in the Cumaru watershed in the "Zona Bragantina," one of the oldest settlement regions of the eastern Amazon. The research strategy was to monitor different ways to manage and prepare land using fallow vegetation: (1) a catchment under mechanical mulching; (2) a catchment under traditional use of fire; and (3) a control catchment covered by secondary forest. The authors found that the catchment where mulch was used performed similarly to the control, in that (a) water balance for the mulch and control catchments was closed on an annual basis; (b) transport of water and nutrients from overland flow or subsurface stormflow was absent; (c), both the mulched and the control catchments showed similar nutrient budgets; and (d) the overall hydrological response was similar [20].

Vegetation in riparian areas, especially wetlands, also plays a key role in nutrient and water flow management, particularly for baseflows and stormflow generation. This riparian vegetation mitigates negative farming impacts on water resources, such as reduced dissolved oxygen concentrations and increased temperature, pH, and electrical conductivity [15,20,21]. Because of their important role in the mitigation or generation of stormflow, such vegetation deserves to remain conserved in order to minimize erosion and sedimentation problems, although further investigation is needed into stormflow-generating mechanisms for streams throughout the Amazon region. The role of riparian forests in securing dry season flows also deserves some attention [22]. In addition, these streams and their riparian areas are important parts of the carbon cycle in the Amazon Basin, as small streams can evade substantial quantities of $CO_2$, especially when compared to fluxes measured for major Amazonian rivers, and thereby serve as an important carbon loss process by rainforests even beyond farming activities [23,24]. Moreover, riparian forests have been shown to benefit aquatic habitat, including for stream fish diversity [25] and for overall water quality for residents, who use water from springs and small streams for household and domestic uses or who may use rudimentary cesspits in the absence of sanitary sewers as this vegetation can capture these materials before they enter stream water [15].

Overall, management of smallholder farms can play an important role in hydrobiogeochemical cycling, aquatic ecosystem conservation, soil conservation, and mitigation of trace gases emissions from biomass burning, especially in small catchments. There are clear, positive differences in (1) water quality indicators between traditionally managed and fire-free managed watersheds and (2) between small landholder versus large-scale producer-managed watersheds [26]; there is, therefore, an opportunity to improve watershed service provisions through ecosystem service bundling that can reward Amazonian smallholders. Based on this knowledge, we present in Figure 2 a conceptual model about the factors and processes associated with impacts by small farming on the biogeochemistry of Amazonian watersheds.

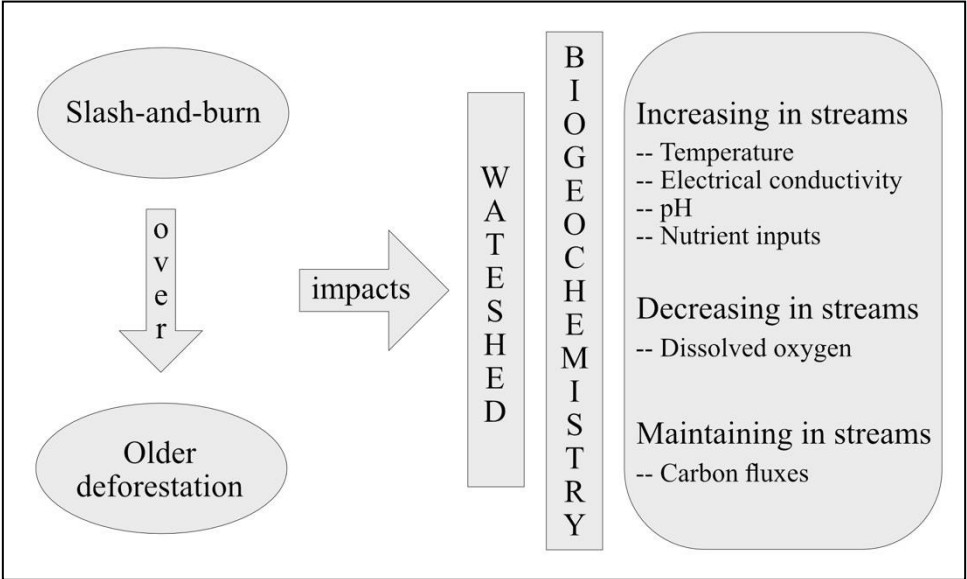

**Figure 2.** Conceptual model of watershed biogeochemistry affected by small farming.

*2.3. Watershed Biogeochemistry Affected by Large Crops and Pasture*

After deforestation for logging or forest burning, conversion of forest to large pasture and crop areas has historically been the main type of land cover change in the Amazon. Several studies have identified how such changes can affect the stream water of small watersheds, though effects can vary locally and regionally depending on variation in soil chemical properties [11]. Moreover, to assess the effects of land use changes on hydrology, it is necessary to understand not only nutrient cycling in the terrestrial component of the catchment but also the interactions between rainfall and physical soil characteristics, which determine the flow paths that will be activated in response to a given rainfall event [27,28]. As such, demonstrating changes in hydrogeochemistry is complex as well as variable on the diverse Amazonian landscapes.

Analyzing data of 10 small watersheds throughout the Amazon, Neill et al. [29] concluded that the contribution of overland flow as a source of stream flow was greater in pasture watersheds than in forest watersheds of comparable size. Large pasture areas, with narrow or nonexistent riparian forests, can alter the chemical composition of rivers, particularly for streamwater cation concentrations and alkalinity [30,31]. As a result of increased root and microbial respiration during the wet season, generated $CO_2$ in soils drives cation-bicarbonate leaching from the landscape to streams and stream water. It is assumed that cations (derived from ashes of forest burning for pasture conversion) and bicarbonates are mainly flushed from surface soil layers by rainfall rather than as the products of deep soil weathering carried by groundwater, with cation concentrations increasing during larger discharge periods [30,31].

Other studies have also found that changes in land cover have larger impacts on stream water chemistry. For example, Markewitz et al. [32] found that soils and land use as well as stream order

all explained portions of the variance in mean calcium concentrations in stream water, while season, year, and vegetative cover explained much of the variance in the slope of the discharge-concentration regression, concluding that land use plays a larger role for discharge-concentration slopes than soil classes. Such results have been shown for other chemical ions in streams in different parts of the Amazon Basin, particularly those draining both large pastures and areas of forest conversion, including chloride, potassium, sodium, sulfate, and overall solutes [32–38]. These changes have also been especially pronounced in the dry season across the Amazon, with one study [39] finding that chloride concentrations increased nonlinearly with deforestation in the dry season. This impact, however, was only observed in watersheds that were more than 66%–75% deforested, suggesting that stream nutrient concentrations are resistant to perturbation from vegetation conversion below this deforestation rate [33]. Importantly, in pastures, salt supplements for cattle may serve as a key source of sodium and chloride to stream water as these salts leach from the landscape [33,34]. Cattle excrement may also serve as an additional source of ions, particularly nitrate [35]. Generally, losses of these ions (with the exception of nitrate, which has been found to have variable flowpaths as discussed below) have been from quickflow or surface flow rather than groundwater outflow. Additionally, conversion of forest to pasture has a lasting, legacy effect on streams; pasture areas have been found to continue to lose important cations even decades after deforestation and pasture establishment [35,40].

Pasture has also impacted the cycling of nitrogen (N), phosphorus (P), organic matter, and suspended solids in stream water. Similar to the ions mentioned above, many of these results have been more pronounced in the dry season [39] and have been found to increase non-linearly with deforestation [33]. Wet season responses were generally diluted (e.g., for total dissolved phosphorus, TDP), or not found (e.g., for particulate phosphorus, PP, or total dissolved nitrogen, TDN) [38]. Compared to forest streams, pasture streams frequently had lower nitrate concentrations but higher dissolved organic nitrogen and phosphorus, leading to small differences in total nitrogen concentrations between land uses; this result suggests that streams switch from phosphorus limitation in forested areas to nitrogen limitation in pasture areas due to lower ratios of inorganic and total dissolved N:P, thereby affecting the structure of these aquatic ecosystems [39,41]. This switch in the system from being dominated by processes producing and consuming nitrate to being dominated by ammonium is presumably the product of lower rates of net nitrogen mineralization and net nitrification in pasture compared to forest [41]. Consequently, in pasture, no hydrological flowpaths contain substantial amounts of nitrate compared to forest catchments. If pasture agriculture in the Amazon shifts toward intensive crop cultivation, this scenario may change; Figueiredo et al. [34] found decreasing nitrate concentrations as forest cover declined and pasture cover increased from upstream to downstream areas, except where crops were grown near the stream, which was associated with increased stream nitrate, likely related to fertilizers. Moreover, agricultural inputs were suspected of promoting not only nitrate spikes but also the observed dissolved oxygen decline.

For P, Hortonian overland flow production is a significant pathway for loss, particularly from mature pasture systems [40]. Similar to other chemical ions described above, phosphorus leaching in pasture can also have a legacy effect on stream systems, with effects continuing for decades following deforestation for pasture [40]. Though most changes have been found in the dry season, phosphorus and other nutrient inputs to streams during the rainy season can be magnified by the fact that deforestation for pasture alters fundamental mechanisms of stormflow generation and may increase runoff volumes over wide regions of the Amazon [42]. As more rainfall in established pastures is diverted into fast flowpaths and to streams, the potential to deliver water with high solute concentrations generated by erosion or by bypassing sites of solute removal increases [43].

In contrast to the above results, in the southeastern Amazon where expansion and intensification of soybean agriculture has occurred on deep and highly permeable soils on broad, upland plateaus small soybean watersheds appear buffered against increased stormflows [14]. In this situation these authors found that concentrations of nitrate and phosphate did not differ between forest or soybean watersheds because fixation of phosphorus fertilizer by iron and aluminum oxides and anion exchange

of nitrate in deep soils restrict nutrient movement. In this same region, despite water yield increasing four-fold from croplands, streamwater chemistry remained largely unchanged [44]. Based on studies from temperate croplands with similar cropping and fertilizer application practices, it is surprising that such low phosphorus and nitrogen stream water fluxes were found.

Despite this limited change in N and P, headwater streams in Amazonian soybean watersheds are experiencing higher temperatures due to farm impoundments and reductions in riparian forest cover [14]. Analyzing 12 catchments for the 176,000 km$^2$ upper Xingu watershed, Macedo et al. [45] demonstrated that streams in pasture and soybean watersheds were significantly warmer than those in forested watersheds, with average daily maxima over 4 °C higher in pasture and 3 °C higher in soybean areas. These authors suggest that management practices associated with recent agricultural expansion may have already increased headwater stream temperatures across the Xingu.

The relationship of pasture to nearby riparian areas also influences carbon fluxes in Amazonian systems. The carbon flux return from water to the atmosphere in the Amazon is substantial compared to other ecosystem processes (e.g., soil respiration) [23]. In this context, not only are large rivers and flood plains important, but so are small streams and their riparian zones, which provide the largest area for terrestrial to aquatic ecosystem exchange. Carbon fluxes in the Amazon are supported largely by terrestrial carbon entering streams and rivers as groundwater $CO_2$ or as terrestrial organic matter that was subsequently mineralized by the aquatic ecosystem. This cycling can be highly affected by land use change; thus, continued study is necessary to improve our understanding of how and how strongly Amazonian ecosystems act as sinks or sources of carbon, particularly through the amount of C that is carried downstream [46]. Davidson et al. [47], in a study measuring $pCO_2$ at multiple stations along three streams from their headwaters in remnant mature forests through multiple land uses in the eastern Amazon, found that downstream $pCO_2$ values appear to be in a quasi-steady state, indicating important contributions from other carbon sources, such as aquatic primary production, soil erosion, dissolved organic matter, or litter inputs from streamside vegetation. Hence, lateral $pCO_2$ loss from groundwater to streams may be minor for most of the terrestrial ecosystems in the Amazon, although carbon loss to streams could be significant for terrestrial budgets in riparian ecosystems or areas experiencing erosion.

Another pathway for carbon loss in the Amazon is through fluxes of dissolved organic carbon (DOC), particularly during the dry season when decomposition of allochthonous litterfall is likely the primary factor for in-stream generation of DOC [48]. Significant in-stream processing of terrestrially derived carbon also has been shown to occur in headwater streams, with particulate organic carbon largely spiraled downstream by stormflow, while DOC was more constantly mobilized from the headwater catchments [48]. As the organic carbon transported by headwater streams to larger streams and rivers undergoes both terrestrial and aquatic processing within the headwater catchments, we can assume that deforestation can disrupt such regional carbon cycling as well as other biogeochemical cycles.

While mostly studied in small watershed catchments in the Amazon Basin, these patterns have also been found at larger scales; for example, one study looked at the biogeochemistry of surface water of one Amazon River tributary and found that areas dominated by pasture at the large-scale corresponded with surface water containing higher concentrations of all studied ions ($Na^+$, $Ca^{2+}$, $Mg^{2+}$, $K^+$, $Cl^-$ and $PO_4^{3-}$), with the highest concentrations found in the central part of the watershed where pasture areas were at a maximum [36]. Concentrations of these ions in stream water predictably dropped in the lower reaches where the river entered areas that were more heavily forested. In the same river basin, however, land use was found to disrupt ecosystem structure and function at the micro- and meso-scales, with significant alterations to nutrient cycles in fluvial ecosystems, though this was not detected at the macro-scale of the entire Amazon basin [37]. In another study, Thomas et al. [10] observed a strong influence of land use across stream order and size in the southwestern Brazilian Amazon, noting that as a second order stream exited forest and entered pasture, concentrations of dissolved oxygen dropped from 6 mg L$^{-1}$ to almost 0 mg L$^{-1}$ and nitrate concentrations dropped

from 12 μM to 2 μM over a reach of 2 km. Similarly, urbanization, coupled with deforestation, has played an important role in the biogeochemistry of Amazonian streams and rivers, including changes to stream water chloride, nitrogen, and phosphorus concentrations at a wide range of scales, from small pasture streams to large river systems [15,38]. Based on this knowledge we present in Figure 3 a conceptual model about the factors and processes associated with the impacts of pasture and large-scale agriculture on the biogeochemistry of Amazonian watersheds.

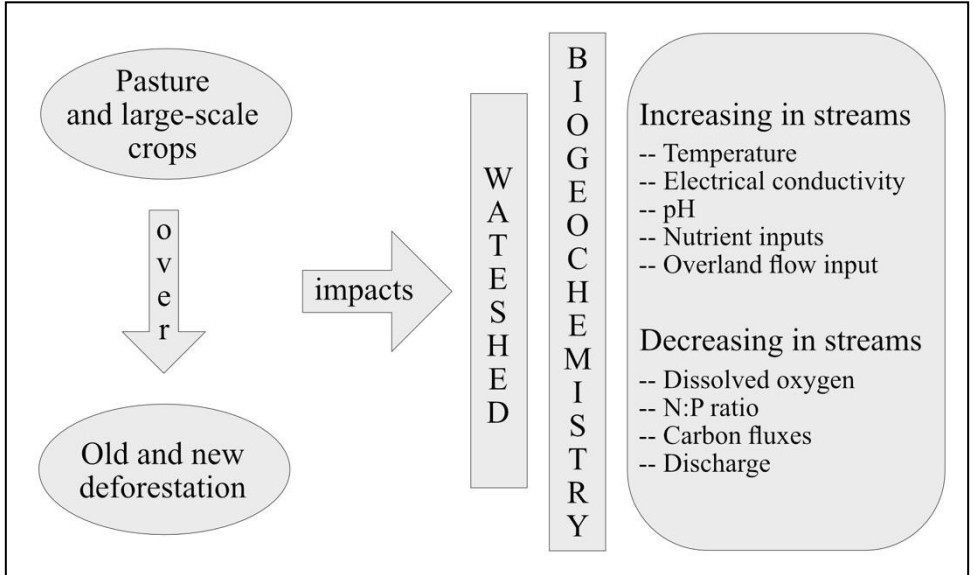

**Figure 3.** Conceptual model of watershed biogeochemistry affected by large crops and pasture.

## 3. Solutions

### 3.1. Agro-productivity and Alternative Agro-systems

In an increasingly globalized world, the key challenge for agriculture is to increase food production from existing farmland in ways that place far less pressure on the environment [49]. One proposed method by Garnett et al. [49] is a "sustainable intensification" (SI) approach as a policy goal for many institutions, though this approach has been criticized for being too narrowly focused on production or being a fundamental contradiction in terms. However, as they note, SI is only a single part of what is needed to improve food system sustainability and is by no means synonymous with food security. SI denotes a goal but does not specify *a priori* how it should be attained or which agricultural techniques should be deployed. Among several approaches that merit further testing and assessment are conventional, "high tech," agro-ecological, or organic, each of which needs to consider biophysical and social contexts.

For the Amazon, several potential strategies are being or can be applied in small farming and/or agribusiness. For example, the no-till system (NTS) is one potential option for a more sustainable form of intensification. It can be understood as a complex of technological processes intended for the exploitation of productive agricultural systems, comprising soil tillage only in the sowing row or in the seeding pit, as well as a permanent maintenance of land cover and species diversification via crop rotation and/or intercropping [50,51]. While contributing to positive impacts on watershed biogeochemistry, NTS, given its management approach that includes the use of herbicides, for example, has been blamed for changes in weed biodiversity, disease inoculums, as well as changes in insect occurrence [52].

The adoption of a no-till system (NTS) combined with crop-livestock integration (CLI) in Brazil is another alternative strategy that has been promoted for sustainable intensification. Silva et al. [51] concluded in an experiment that grazing did not alter the yield of soybeans, while pasture and

animals did not affect corn plant population attributes and yield or the percentage of damaged kernels; furthermore, animal grazing during the winter caused increases in the corn production and grain yield. This approach can be a solution, with several successful examples of NTS with CLI occurring in the Paragominas region of the eastern Brazilian Amazon [53]. These combined approaches can be disseminated and multiplied with public and private support, not just in this region of Brazil, but the Amazon more broadly, though such a policy goal requires an increased need for scientific study of this productive system, particularly with regard to stream water benefits.

Another alternative for sustainable agriculture is the use of multistrata agroforestry systems. Such an approach has several advantages over monocultures as it combines high and long-term biomass accumulation with early generation of income from annual and semi-perennial intercrops, increased growth and earlier yields of certain tree crops, long-term accumulation of capital in larger trees, and more complete occupation of the soil than in common tree crop monocultures of the Amazon region [54]. Agroforestry has been successfully used for restoration of degraded (or altered) areas in order to reconcile environmental conservation with social and economic benefits, among which include water environmental services in Brazil [55].

Furthermore, as noted earlier, a chop-and-mulching system can serve to help conserve soil moisture, reduce soil erosion, and attenuate the entry of agrochemicals in the hydrochemical flows of the basin [56]. Cutting fallow vegetation (secondary forest, called "capoeira" in Brazil) followed by shredding of this cleared vegetation can be used in place of fire to prepare the land for agriculture, as fire has negative impacts on soil, water, and air quality [18]. Results from two decades of research have shown this practice to have good result for agricultural production outcomes, together with conservation of soils and water resources [57].

Non-Timber Forest Products (NTFPs) can also play a role for sustainable agriculture in the Amazon as well as an additional alternative for the rural economy. There is no evidence and little reason to expect that harvest of NTFP will have any negative impact on stream waters. In particular, there are three important NTFPs in the Amazon that contribute to the livelihoods of more than 6 million households in the Brazilian Amazon: rubber, Brazil nut, and açaí [58]. Açaí, for example, has a great worldwide market potential as a healthy "superfood" around the planet [59]. These extractivist activities also can be reconciled with a broad set of agricultural activities for a diverse agricultural portfolio. Many other products from the forest also have medicinal potential and can serve as important products for a more sustainable use of the Amazon; for example, there are a lot of species and genes that encode enzymes involved in the complex metabolism of organisms, a rich chemical diversity, which is a potential source for a bioeconomy based on natural products and new synthetic derivatives [60].

### 3.2. Tradeoffs and Public Policies

Nowadays the Amazon is not merely a frontier, but a thriving region in itself, and one that demands a development policy. The forest will only be protected if it has social virtues and economic value that can compete with wood, cattle, and soy, particularly by maintaining the sustainable capacity of the forest. There is a need to establish biodiversity-based techno-productive chains, from forest communities to the centers of advanced technology [61]. While much research over the past few decades has increased our understanding of ecosystem processes in the Amazon, much work still remains to further our knowledge regarding development plans for the region that integrates forest conservation with the needs of Amazonian residents. This can be undertaken with the development of new and more advanced sensors that can be built and deployed (often at low-cost) to measure the environment across broad scales and remotely [62,63], as well as through use of new computational advances such as artificial intelligence and machine learning to analyze these enormous datasets for further insights that may otherwise not be detected from current statistical approaches [64].

For Amazonian watershed management, the adoption of payment for environmental services (PES) stands out today as an intriguing way to meet the aforementioned challenges, as this can be an important instrument for the implementation of actions related to hydro-environmental recovery [65]. For example,

for landowners to receive payments for watershed conservation, and its related hydrobiogeochemistry functions, they may be required to adopt soil conservation and reforestation practices near springs and other water sources, as well as along stream and river riparian margins.

Furthermore, the program to register property boundaries not just in Brazil but across the Amazon has the potential to help farmers and landowners to stay in compliance with the law, in Brazil known as "Cadastro Ambiental Rural—CAR" or the Rural Environmental Registry [66]. During field research, farmers frequently mentioned to one of the co-authors that they would only be happy to be in compliance with the law (e.g., the requirement to maintain unused forested areas or riparian buffers through the "Permanent Preservation Areas"—PPAs) if they were able to adequately register their property locations. Such registration can help to resolve complicated legal boundaries, such as the location of these PPAs in relation to other forested areas or other areas of use.

## 4. Conclusions

Studies over the past 20 years point out that for both property types—smallholders and large farmers—riparian forest conservation can make a significant difference in reducing impacts of land-use change on biogeochemical processes at the different watershed scales in the Amazon. Additionally, there is evidence that secondary vegetation can also play an important role in both mitigating these impacts and helping to maintain and support water quality and stream ecosystem functions.

Additionally, new incentives for agricultural production, such as payments for ecosystem services, together with new advances in agricultural production and alternative agro-systems, can contribute to healthy ecosystems. The agricultural solutions presented in this article can be part of a number of recommendations for land management policy that address tradeoffs among food production, forests, and water resources in the agriculture frontiers of the Amazon. These sustainable development plans, together with continued study of these systems and the impacts of these plans, can help to assure that forests remain conserved while the needs of Amazonian riverine inhabitants are met.

**Supplementary Materials:** The following are available online at http://www.mdpi.com/2073-4441/12/3/763/s1, Publications used to generate Figure 1.

**Author Contributions:** Conceptualization, R.d.O.F. and A.C.; methodology, R.d.O.F. and A.C.; formal analysis, A.C.; data curation, R.d.O.F. and A.C.; writing—original draft preparation, R.d.O.F., A.C. and D.M.; writing—review and editing, R.d.O.F., A.C. and D.M.; visualization, R.d.O.F. and A.C.; supervision, R.d.O.F. All authors have read and agree to the published version of the manuscript.

**Funding:** This research received no external funding.

**Acknowledgments:** The authors are grateful for all research teams that developed the studies cited in this review paper.

**Conflicts of Interest:** The authors declare no conflict of interest.

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
