# Peer review of "Agricultural Impacts on Hydrobiogeochemical Cycling in the Amazon: Is There Any Solution?"

_water, doi:10.3390/w12030763_

Round 1

Reviewer 1 Report

This review paper is really interesting. It provides a very detailed collection of papers that focus on agricultural Impacts on the Hydrobiogeochemical Cycling in the Amazon, while it also includes intersting aspects related to solutions of relevant issues. The paper can be accepted after minor revision of typo and language errors (see attached file). I recommend the authors to make a final reading of their paper for laguage improvements.

Author Response

Point 1: The paper can be accepted after minor revision of typo and language errors (see attached file).

Response 1: All revision of typo and language were done as suggested.

Point 2: I recommend the authors to make a final reading of their paper for language improvements.

Response 2: We have reedited the manuscript for language improvements as recommended.

Reviewer 2 Report

The manuscript "Agricultural Impacts on the Hydrobiogeochemical Cycling in the Amazon: Is there any solution?" is an important contribution to the specific field, and more. While this is an important piece, it is still difficult to capture the most important points in the text due to lack of clarity, coherence, and conciseness. This can be improved by MAJOR revisions throughout the text. I have left several comments, but the authors should dedicate some time to improve the readability of the entire text. The Abstract must be very clear to a large audience, as it is a way to engaje the reader and increase the number of citations. Also, the Conclusion is poorly written and does not make it clear what is the main contribution of this study. Throughout the text, try to minimize clauses including "XY found this...", "Z found that...". Rather, summarize the finding and include the citation at the end. I understand this manuscript presents a summary of findings, but the reader will benefit from having shorter sentences with the information summarized.

Author Response

Point 1: Comments in the manuscript regarding to English usage and grammar.

Response 1: Most of the revisions were done as suggested but not all as we understand the use of a few phrases in other ways.

Point 2: Throughout the text, try to minimize clauses including "XY found this...", "Z found that...".

Response 2: We have done this as suggested.

Point 3: It is still difficult to capture the most important points in the text due to lack of clarity, coherence, and conciseness.

Response 3: To follow such recommendation we have written many portions of the manuscript, mainly in its the second half. Also we cut off one reference paper regarding to pesticides as this does not match well with our discussion. Due to this and the new text the reference list now has been changed in terms of order.

Round 2

Reviewer 2 Report

I reviewed the manuscript "Agricultural Impacts on Hydrobiogeochemical Cycling in the Amazon: Is there any solution?" for the second time and appreciate the authors' effort in improving the quality of the manuscript. I only have a few minor comments (attached), which should be easily addressed by the authors. I do recommend this manuscript for publication after those comments are addressed. 

Author Response

Jaguariúna, February 26, 2020

To Water Editors

Cover Letter

The manuscript "Agricultural Impacts on the Hydrobiogeochemical Cycling in the Amazon: Is there any solution?" was reviewed according reviewers comments as follow:

line 17 - We have inserted the phrase "for cultivation" after "prepare land" to clarify what "prepare land" means as asked by the reviewer.

line 17 - We have changed the phrase "a problem" to "controversial" as suggested by the reviewer.

line 39 - We have changed the phrase "in the Amazon, and Brazil, in particular" to "in the Brazilian Amazon" as suggested by the reviewer.

line 43 - We have removed the words "have been shown to" as suggested by the reviewer so that the phrase now reads "small farmers contribute".

line 127 - We removed the word "Brazil" as suggested by the reviewer.

line 133 - We have spelled out the acronym "DO" to "dissolved oxygen" as suggested by the reviewer.

line 137 - We have clarified which are the nutrients provided by throughfall and stemflow.

line 158 - We have kept "dissolved oxygen" as it is the acronym "DO" used previously.

line 244 - We have kept "P" instead of "phosphorus" because it is the way this nutrient is mentioned in the previous paragraph.

line 265 - We have included the word "cover" after "riparian forest" as suggested by the reviewer.

line 272 - We have corrected the phrase "The carbon flux return from water" as suggested by the reviewer.

line 378 - We have corrected the subtitle of this item 3.2 to "Tradeoffs and public policies". We do not know how in the previous versions of this manuscript the title of item 3.1 was repeated as the title of 3.2.

Sincerely,

Ricardo Figueiredo